# Tibial Tubercle to Trochlear Groove Distance Measured by Posterior Condylar Reference Line on MRI Can Over-Evaluate Lateralization Deformity of Tibial Tubercle in Patients with Recurrent Patellar Dislocation

**DOI:** 10.3390/jcm11175072

**Published:** 2022-08-29

**Authors:** Pei Zhao, Jiaxing Chen, Yi Feng, Hao Tan, Baoshan Yin, Hua Zhang, Jian Zhang, Aiguo Zhou

**Affiliations:** 1Department of Orthopedics, The First Affiliated Hospital of Chongqing Medical University, Chongqing 400016, China; 2Medical Education Department, The First Affiliated Hospital of Chongqing Medical University, Chongqing 400016, China

**Keywords:** patellar dislocation, tibial tubercle to trochlear groove distance, transepicondylar axis, tibial tubercle osteotomy, MRI

## Abstract

Background: The tibial tubercle to trochlear groove (TT-TG) distance is currently considered as an indication for tibial tubercle osteotomy. While the influence of femoral condylar morphology on such measurement remains unclear. Methods: A total of 86 patients with patellar dislocation (PD) and 86 healthy individuals were enrolled. Femoral condylar morphology and the TT-TG distance measured by anatomical transepicondylar axis (TT-TGa), by surgical transepicondylar axis (TT-TGs), and by posterior condylar reference line (TT-TGp) were assessed by MRI. Unpaired t-test, Spearman, and Pearson correlation analysis were conducted. We determined the pathological value of the parameters and established a binary regression model. Results: The interclass correlation coefficients of all the TT-TG distances were greater than 0.75 in all types of trochlear dysplasia. The lateral/posterior femoral condyle was shorter and the medial/posterior condyle was longer in the study group (28.5 ± 3.3 and 35.2 ± 2.8, respectively) than in the control group (30.9 ± 2.7 and 33.5 ± 2.3, respectively). In the study group, the TT-TGp distance was greater than TT-TGs and TT-TGa distance (*p* < 0.001). The pathological value of the TT-TG distance was 13.0 mm. Each TT-TG distance revealed a significant OR with regard to PD. Conclusion: The TT-TGa, TT-TGs, and TT-TGp distance can be reliably measured by MRI even in patients with trochlear dysplasia. While the TT-TGp distance may overestimate the lateralization deformity of the tibial tubercle. Posterior femoral condylar dysplasia may be a reason for such overestimating. These findings have not been correlated to clinical outcomes and further studies are required.

## 1. Introduction

Patellar dislocation (PD) is a common disease of sports injuries [1]. Patients with anatomical variations of the femur and tibia are prone to suffer from PD [2,3,4]. Among them, the tibial tubercle to trochlear groove (TT-TG) distance has been accepted as a significant risk factor, increasing lateral vector applied to the patella [5]. In addition, a large proportion of patients with PD are accompanied by trochlear dysplasia and/or excessive femoral anteversion [6,7]. The TT-TG distance was significantly correlated with other anatomical parameters in patients with PD, such as patella alta [8,9].

The value of the TT-TG distance more than 20 mm is considered an indication for the tibial tubercle medialization osteotomy, and if patella alta appears simultaneously, the tibial tubercle distalization should also be considered (TTO) [10]. But the reliability of the TT-TG distance measured by computed tomography (CT) images in patients with PD is questioned. The femoral trochlear morphology distinctly influences the measurement of the TT-TG distance and it is difficult to identify the deepest portion of the femoral trochlea, even the trochlea itself in patients with severe trochlear dysplasia [11,12]. On the other hand, such measurement was verified reliable on axial magnetic resonance imaging (MRI) cuts [13,14].

The tibial tubercle to posterior cruciate ligament (TT-PCL) distance and the tibial tubercle to Roman arch (TT-RA) distance were depicted by researchers because of their independence from trochlear dysplasia [15,16]. The tangent line from the posterior bony or cartilaginous edge of the medial to the lateral condyle (PCRL) was the most commonly used reference line in the evaluation of tibial tubercle lateralization. However, patients with PD are often in the presence of dysplastic posterior condyles: a shorter posterior lateral condyle and a longer posterior medial condyle [17,18]. Theoretically, the posterior condylar dysplasia could make more internal rotations in the PCRL, influencing the authenticity of the recorded TT-TG distance (TT-TGp). Previous studies were aimed at the influence of the femoral trochlea on the evaluation of tibial tubercle lateralization but ignored the effect of the PCRL as the reference line on the measurement of the TT-TG distance.

The transepicondylar axis has been proven to be a reliable and constant reference line on the distal femoral condyles [19], consisting of anatomical and surgical transepicondylar axis (AEA and SEA, respectively). Both of them have been poorly studied in patients with PD [17,20,21,22,23]. We first proposed to measure the TT-TG distance using AEA and SEA as reference lines on axial MRI slices (TT-TGa and TT-TGs, respectively), which were theoretically independent of posterior femoral condylar dysplasia. The hypothesis of our study is that the TT-TG distance can be reliably measured by PCRL or transepicondylar axis on MRI and the TT-TGp distance differs from the TT-TTa or the TT-TGs distance in patients with femoral condyle dysplasia. In 2022, a systematic review suggested that an individualized surgical strategy should be selected according to the specific etiology of the patients with regard to patellofemoral joints, which lies on a relevant radiological investigation [24]. Thus, it is important to accurately assess the specific anatomic abnormalities in patients with PD.

Given the insufficient knowledge in the literature, the purpose of this study was (1) to verify the reliability of the TT-TG distance measured by PCRL, SEA, and AEA on MRI, (2) to validate the diagnostic capacity of each parameter for PD, (3) to define the difference among the three TT-TG distances in patients with PD.

## 2. Materials and Methods

### 2.1. Study Population

This study was approved by the ethics committee of our hospital. Patients diagnosed with PD from 2016 to 2021 in our hospital were selected from the Electronic Medical Record System. The inclusion and exclusion processes were conducted by two experienced orthopedic surgeons. A total of 106 patients were considered eligible for inclusion in this study according to the inclusion criteria: patients diagnosed with unilateral recurrent PD; patients with skeletal maturity; patients with MRI scans of the ipsilateral knees. Patients who met at least one of the following criteria were excluded: patients missing standard or clear MRI scans of the knees (n = 15); patients with a history of bone fracture or surgery that may influence the reliability of the measurements (n = 3); patients with severe epiphysitis of the femur (n = 2). Consequently, 86 patients were designated as the study group.

Then, 86 individuals matched by age and sex with knee MRI showing neither pathology, meniscal nor cruciate ligament injury were assigned as control group in this study. The informed consent was waived by the hospital ethics committee. Medical history inquiries of all the individuals in the control group were conducted by a junior orthopedist and the MRI were evaluated by a senior orthopedist to ensure their eligibility for this study. None of these subjects had symptoms or objective pathologic findings related to the patellofemoral joints.

### 2.2. MRI Technique

The MRI examinations were performed within one week prior to surgical planning in patients with PD. All examinations were performed with the same 1.5T MRI scanner (Siemens Magnetom Essenza, Munich, Germany). All the participants were placed in the supine position with the leg in full extension and the foot in a neutral position. The target knee was secured in the multichannel 89 phased-array knee coil, and the scan range was centered on the tibia and included the entire knee joint. The coronal and sagittal planes were scanned with the T1-weighted turbo spin-echo (TSE) sequence and proton density (PD) TSE with the fat-suppressed (FS) sequence, and the axial plane was scanned with the PD-TSE-FS. The layer thickness was set to 4 mm, the slice gap was 0.5 mm, the field of view (FOV) was 160 mm, and the matrix size was 512 × 512.

### 2.3. Measurements of Parameters

All the parameters were assessed on MRI slices by a senior orthopedist and a junior orthopedist in a blinded and randomized fashion using the picture archiving and communication system (PACS). To standardize the measurements, the observers met as a group and agreed upon the measuring methods. If a major dispute of the measuring result was presented, especially the degree of trochlear dysplasia, a discussion would be conducted until consensus was reached.

#### 2.3.1. TT-TG Distance

The TT-TG distance using PCRL as the reference line was measured on the MRI scan according to the method described by Friedman et al. [25]. The most proximal axial slice showing the complete femoral trochlea with cartilage and intact posterior femoral condyles was identified to define the following marks: the deepest portion of the cartilaginous trochlear groove, the tangent line from the posterior cartilaginous edge of the medial to the lateral condyle (posterior condylar reference line, PCRL), the line through the sulcus of the medial epicondyle and the prominence of the lateral epicondyle (SEA) [26], and the line through the prominence of the medial and lateral epicondyles (AEA) [27] (Figure 1A). The first axial image showing a complete patellar tendon attachment on the tibial tubercle was selected as the reference slice to identify the center of the tibial tubercle. Then, the TT-TG distances were evaluated: TT-TGp distance (Figure 1B) was defined as the distance between two perpendicular lines of the PCRL (the one across the midpoint of the superficial aspect of the tibial tubercle and the other one passing the deepest point of the trochlear groove); likewise, the TT-TGs distance (Figure 1C) and TT-TGa distance (Figure 1D) were defined as the distance between two vertical lines of the SEA and two vertical lines of the AEA, respectively.

#### 2.3.2. Posterior Femoral Condylar Morphology

To assess the posterior femoral condylar morphology, we measured the distances between the SEA and the posterior cartilaginous margin of the medial femoral condyle (MPD) and the lateral femoral condyle (LPD) on the same distal reference axial MRI slice used to define the deepest portion of the trochlear groove [19]. The transepicondylar width (TEW) is defined as the distance between the sulcus of the medial epicondyle and the prominence of the lateral epicondyle. The posterior condylar angle (PCA) is the angle between the SEA and PCRL, which is 3 degrees or so in normal population [26] (Figure 2).

#### 2.3.3. Trochlear Dysplasia

Sharma et al. [28] depicted a new classification method to simplify the evaluation of trochlear dysplasia and reported it as a reliable and superior alternative method compared to the Dejour classification. Furthermore, he identified that the Oswestry-Bristol Classification (OBC) played an important role in the surgical decision-making [29]. It is a four-part classification system comprising normal, mild, moderate, and severe trochlear dysplasia. Mild trochlear dysplasia represents a shallow femoral trochlea; moderate trochlear dysplasia represents a flat femoral trochlea; severe means a convex trochlea (Figure 3). The inter-observer reliability of the OBC showed excellent agreement, with the ICC value of 0.869 (95% CI [0.843, 0.889]) via Weighted Kappa analysis.

### 2.4. Statistical Analysis

Another orthopedist conducted all the analysis independently using the average value of the parameters measured by the two observers via the SPSS software (version 21.0; IBM Corp, Armonk, NY, USA). Those length parameters were normalized by TEW to reduce individual differences. The Shapiro–Wilk normality test was performed to verify the distribution of these parameters. Then independent *t*-test and Mann–Whitney U test were conducted to assess the differences in the anatomical parameters between the groups. Pearson and Spearman correlation analyses were used to verify the relationships between these parameters in the study group. Receiver operating characteristic curves (ROC) and the areas under the curves (AUC) were analyzed to evaluate the diagnostic capacity of the parameters for PD via GraphPad Software (version 8.0.2, San Diego, CA, USA). An AUC greater than 0.7 indicates fair to good capacity [14]. The data in the control group were used to identify the pathological threshold value of each TT-TG distance. We established the Binary logistic regression model based on the pathological value of the TT-TG distances to identify the anatomical risk factors for PD. The intraclass correlation coefficients (ICCs) of the TT-TG distances in different degrees of trochlear dysplasia were calculated to identify the reliabilities, with a value more than 0.75 indicating excellent agreement [14]. We defined α as 0.05.

We conducted the Post hoc analysis via G-Power software (version 3.1.9.4, Heinrich-Heine-Universitat Dusseldorf, Dusseldorf, Germany). For the effect size of 1.08 according to the TT-TGs between the two groups, a power of 1.00 was calculated (n, 86; alpha, 0.05).

## 3. Results

This study included 86 patients with PD in the study group and 86 healthy individuals in the control group. The demographic data of the two groups are shown in Table 1. Except for PCA and age, all the parameters conformed to normal distribution.

The TT-TGp distance was greater in the study group than in the control group (12.7 ± 4.0 mm and 8.2 ± 2.9 mm, respectively), and likewise the TT-TGs and TT-TGa distance. The ratios of TT-TG distances/TEW were larger in the study group compared to that in the control group (*p* < 0.001). The LPD/TEW was smaller in patients with PD (28.5 ± 3.3%) compared to that in healthy subjects (30.9 ± 2.7%), in contrast, the MPD/TEW was larger in the study group (35.2 ± 2.8%) compared to controls (33.5 ± 2.3%). Then the ratio of LPD/MPD was 81.3 ± 6.6% in the study group and 92.3 ± 5.2% in the control group (*p* < 0.001). The PCA were larger in patients with PD compared to that in the control group (*p* = 0.005). Trochlear dysplasia was presented in patients with PD: four (4.6%) patients with normal, 24 (27.9%) patients with mild, 20 (23.2%) patients with moderate, and 38 (44.2%) patients with severe trochlear dysplasia (Table 2). In the study group, the TT-TGp was greater than TT-TGs and TT-TGa distance (*p* < 0.001), and the TT-TGp/TEW was greater than TT-TGs/TEW and TT-TGa/TEW (*p* < 0.001) (Figure 4).

Spearman correlation analysis showed that the TT-TGp, TT-TGs, and TT-TGa distance was correlated with trochlear dysplasia in the study group with a ρ of 0.47, 0.46, and 0.50, respectively (*p* < 0.001). But the correlations between the TT-TG parameters and the measurements of femoral condylar morphologies did not show statistical significance (Table 3).

The ROC curves were analyzed to calculate the diagnostic capacity of each TT-TG distance and the ratio of the TT-TG distance/TEW for PD (Table 4). The TT-TGp had an AUC of 0.811, with a value of >9.3 mm indicating 80.2% sensitivity and 67.4% specificity. The TT-TGs had an AUC of 0.777. At a value of more than 8.5 mm, the TT-TGs distance had 82.6% sensitivity and 60.0% specificity for predicting PD. The TT-TGa had an AUC of 0.768, with a cutoff value of 9.4 mm (sensitivity 68.6%, and specificity 70.9%) (Figure 5A). The ROC curves of the TT-TG distances/TEW were displayed in Figure 5B.

The pathological threshold value of the TT-TG distances and the ratio of the TT-TG/TEW was calculated according to the mean and standard deviation in the control group (mean + 1.645 standard deviation, Table 5). 47.7% (41/86) patients with PD had a value of the TT-TGp distance greater than 12.9 mm, 40.7% (35/86) patients had a value of the TT-TGs distance more than 13.0 mm, and 39.5% (34/86) patients had a value of the TT-TGa distance greater than 12.9 mm in the study group. The results of Binary logistic regression analysis are shown in Table 5, as eliminating other parameters from the regression model (simple analysis), the TT-TGp, TT-TGs, and TT-TGa distance revealed a significant OR of 18.68, OR of 11.12, and OR of 14.76 with regard to PD, respectively.

Furthermore, we calculated the ICC value to identify the inter-observer reliability of the TT-TG distances in different types of trochlear dysplasia. The ICC of the three TT-TG distances were more than 0.85 in any type of OBC (Table 6). The value of each TT-TG distance was maximum in patients with severe trochlear dysplasia and was minimum in subjects with normal trochlea. The TT-TG distance in patients with shallow trochlea did not differ from that in patients with flat trochlea (Figure 6).

## 4. Discussion

The most important finding of the present study is that the TT-TGa, TT-TGs, and TT-TGp distances can be reliably measured by MRI, even in patients with severer trochlear dysplasia. Each TT-TG distance has fair to good diagnostic ability for predicting PD. In patients with PD, the TT-TG distance is largely measured by PCRL than measured by transepicondylar axis, while such difference is not significant in healthy individuals, indicating that previously recorded TT-TG distance may overestimate the deformity of tibial tubercle lateralization. The pathological value of each TT-TG distance has been identified, which could help surgeons with surgical decision-making.

As an important risk factor for affecting patellar stability, the TT-TG distance can indirectly reflect the lateralization of the tibial tubercle, with a value greater than 20 mm indicating pathological value [8]. The TT-TG distance measured by CT images was unreliable, because the measurement was greatly affected by dysplastic femoral trochlea [12]. On the other hand, previous literature suggested that the TT-TG distance can be reliably measured by MRI, because the cartilage of femoral trochlea can be clearly visualized and the entry of the trochlea can be accurately located by MRI [14,25]. The results of our study showed that all the three TT-TG distances measured by MRI had excellent interobserver agreements, even in patients with severe trochlear dysplasia.

Yang et al. [18] suggested that the dysplastic distal femoral condyle was composed of anterior and posterior femoral condylar dysplasia. In accordance with previous studies on CT images [17,20,22], the results of our study by MRI showed that patients with PD was in the presence of posterior femoral condylar dysplasia: longer medial and shorter lateral posterior condyle, and larger PCA. Previous studies focused on identifying a reference point that could avoid the influence of the femoral trochlea on assessing the lateralization of the tibial tubercle in patients with trochlear dysplasia, such as the TT-PCL distance and TT-RA distance [15,16], while ignoring the PCRL—the reference line in most measurements—may be affected by posterior femoral condylar morphology (the posterior condylar dysplasia could make the PCRL more internal rotation). So, we reasoned that the TT-TG distance measured by PCRL could not accurately assess the lateralization of the tibial tubercle in patients with PD.

In this study, we found that the TT-TG distances were larger in patients with PD than in healthy individuals, which was consistent with previous studies [15,30]. In patients with PD, the mean difference between the TT-TGp distance and the TT-TGs or TT-TGa distance was 0.7 mm, but such difference in healthy individuals was not significant. Such results may attribute to the influence of the dysplastic posterior femoral condyles on the measurement of TT-TGp, while the TT-TG distance measured by transepicondylar axis—a reliable and constant reference line—was independent of posterior condylar dysplasia. This to some extent suggests that the TT-TG distance measured by PCRL could over-evaluate the lateralization of the tibial tubercle in patients with dysplastic posterior femoral condyles. Whether this 0.7 mm deviation has an impact on TTO decision-making or on postoperative outcomes is still unknown. All the three TT-TG distances were not statistically correlated with PCA or LPD/MPD in patients with PD based on our data, which may attribute to the posterior condylar morphological parameters included in this study. The relationship between other parameters of posterior condylar morphology (e.g., lateral femoral condyle index and lateral condyle index) [31,32] and the TT-TG distance warrants further investigation.

Trochlear dysplasia was considered as a significant risk factor for PD, which can change the interaction between the patella and the femoral trochlea. Researchers used to evaluate trochlear dysplasia via the Dejour classification and some specific parameters, such as sulcus angle and lateral trochlear facet inclination (LTI) [6]. Dejour classification has been performed on MRI, but poor inter- and intra-observer agreements have been reported [33]. Sharma et al. [28] suggested the OBC to grade the severity of trochlear dysplasia with the superiority of simplicity and good consistency. Our study also confirmed that the four-part classification system was reliable. He also reported that, based on the OBC, 28% patients were classified as “normal” to “mild”, 62% as “moderate” to “severe”. In our study, among the patients with PD, 28 (32.5%) patients have normal to shallow femoral trochlea, 58 (67.5%) patients are in the presence of moderate to severe trochlear dysplasia. A large proportion of patients were accompanied by trochlear dysplasia [6]. In this study we found that all the three TT-TG distances increased with the severity of trochlear dysplasia, which was in accordance with previous studies [9]. It informed the orthopedic surgeons of the existence of excessive TT-TG distance when severe trochlear dysplasia was presented in patients with PD. After normalizing the TT-TG distances by TEW, the ratio of the TT-TG distances/TEW showed similar results.

We also analyzed the diagnostic efficacy of each TT-TG distance for PD and found that the TT-TGp distance had a good diagnostic capacity, with an AUC of 0.811, which was almost equivalent to the results reported by Xu et al., (AUC = 0.820) [14]. While the TT-TGs and TT-TGa distance had a fair capacity, with an AUC of 0.777 and 0.768, respectively. The discrepancies might attribute to the overestimated TT-TGp distance, influenced by the posterior femoral condylar dysplasia. Similar to the previous literature that the normalized TT-TG distance had a stronger ability to predict PD [34,35], the ratios of the TT-TG distances/TEW in our study had stronger diagnostic capacities than the TT-TG distances alone. The TT-TGp distance greater than 20 mm measured on CT images was considered the indication for TTO and good postoperative outcomes following TTO had been reported [10]. However, since the measuring method of the TT-TG distance differs from CT images to MRI slices, the recorded value of the TT-TG distance measured on such two different images are not equivalent in the same patient [14]. So, it is worthy of identifying the pathological threshold value of each TT-TG distance measured on MRI cuts.

We set the pathological value of all the three TT-TG distances measured on MRI slices at 13.0 mm in our study. Moreover, 0.6 mm discrepancy was identified when compared with previous literature (13.6 mm) [14], which may had been caused by the inclusion of the control group. The pathological value of the TT-TG distances/TEW was set at 17.0% based on our data. It does not mean that the TTO is needed when the TT-TG distance or TT-TG distances/TEW manifests pathology. We expected to establish a “data base” regarding the indication for TTO based on MRI slices to help orthopedic surgeons with surgical decision-making. Whether the 13.0 mm of the TT-TG distance was an appropriate indication for TTO and its postoperative outcomes warrant further investigation.

Previous studies have shown that when a pathological TT-TGp distance is presented, the risk of PD would increase 5–14 times [11,14,34,35]. In our study, all the three TT-TG distances were verified risk factors for PD, and the risk was 18.7, 11.1, and 14.8-fold higher in patients with pathological TT-TGp, TT-TGs, and TT-TGa distance, respectively. This discrepancy may have been caused by the different cutoff values and methods used for logistic regression analysis. we set the cutoff value at 13.0 mm and established simple regression model, but previous studies set the pathological value at 13.6 or 20 mm.

We acknowledged some limitations to this study. First, we did not correlate the operation outcomes following TTO with this new pathological threshold value of each TT-TG distance that we established. Second, the subjects in the control group of this study were not completely normal individuals, but they did not have anterior knee pain, patellar instability, or other symptoms in terms of patellofemoral joints. Pathological values calculated from this sample may change in a relatively large healthy population. Third, this study only included one measuring method of evaluating posterior femoral condylar morphologies, other anatomical parameters, such as lateral femoral condyle index and lateral condyle index were not analyzed. Fourth, only two observers measure the anatomical parameters may lead to biases and the power dynamic between the senior and junior orthopedist may result in bias when there are discrepancies, and the intra-observer reliability of each parameter was not analyzed. Fifth, these anatomical parameters were only measured by MRI, CT images and radiographs should also be investigated.

## 5. Conclusions

The TT-TGa, TT-TGs, and TT-TGp can be reliably measured by MRI even in patients with trochlear dysplasia. While the TT-TGp distance may overestimate the lateralization deformities of the tibial tubercle, posterior femoral condylar dysplasia may be a reason for such overestimating. These findings have not been correlated to clinical outcomes and further studies are required.

## Figures and Tables

**Figure 1 jcm-11-05072-f001:**
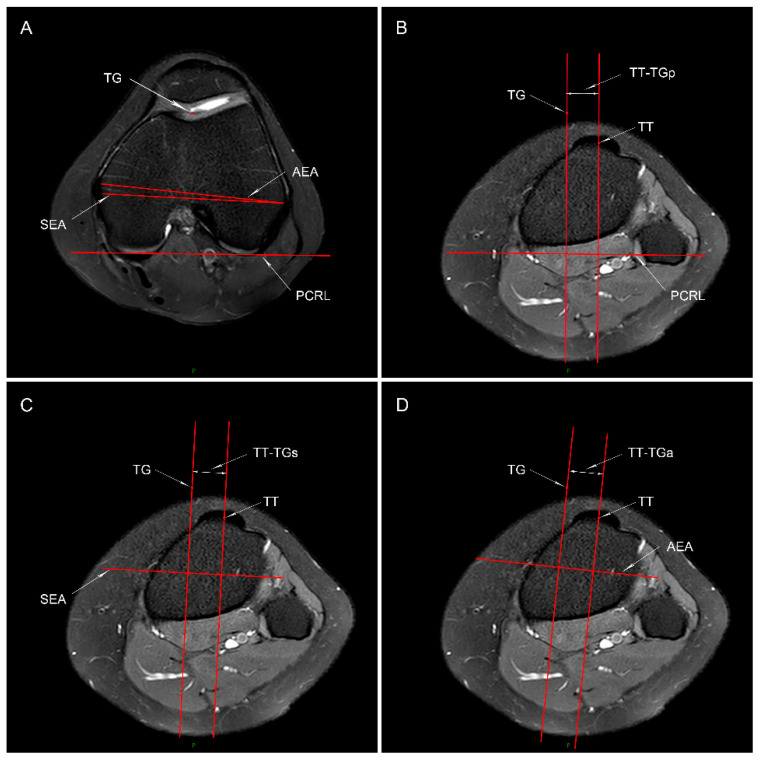
Measuring method of the tibial tubercle-trochlear groove (TT-TG) distance. (**A**) The posterior femoral condylar reference line (PCRL), surgical and anatomical transepicondylar axis (SEA and AEA, respectively), and the deepest portion of the cartilaginous trochlear groove (TG) are shown in the axial MRI plane. (**B**–**D**) The same slice showing the full attachment of the patellar tendon to the tibial tubercle is identified to mark the midpoint of the superficial aspect of tibial tubercle (TT). The TT-TGp (**B**), TT-TGs (**C**), and TT-TGa (**D**) distance is defined as the distance between two lines that are perpendicular to the PCRL, SEA, and AEA, respectively: one through the TG, and the other one through the TT.

**Figure 2 jcm-11-05072-f002:**
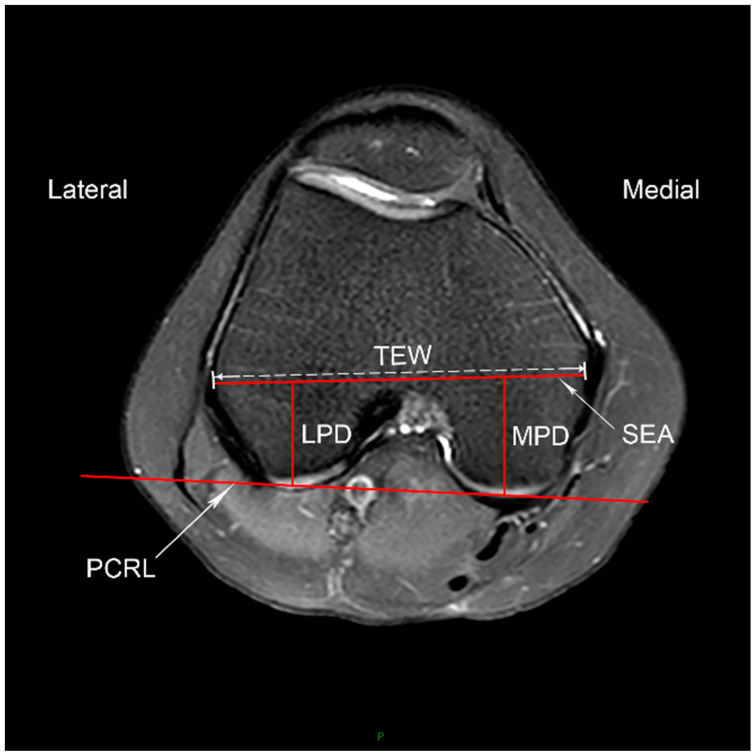
Posterior femoral condylar morphology. The axial MRI slice showing the complete femoral trochlea with cartilage and intact posterior femoral condyles is selected. The posterior femoral condyle reference line (PCRL) and the surgical transepicondylar axis (SEA) are shown. The width of the SEA (TEW), the distance between the SEA and the posterior cartilaginous margin of the lateral condyle (LPD) and medial condyle (MPD) are measured. The posterior condylar angle (PCA) is the angle between the SEA and PCRL.

**Figure 3 jcm-11-05072-f003:**
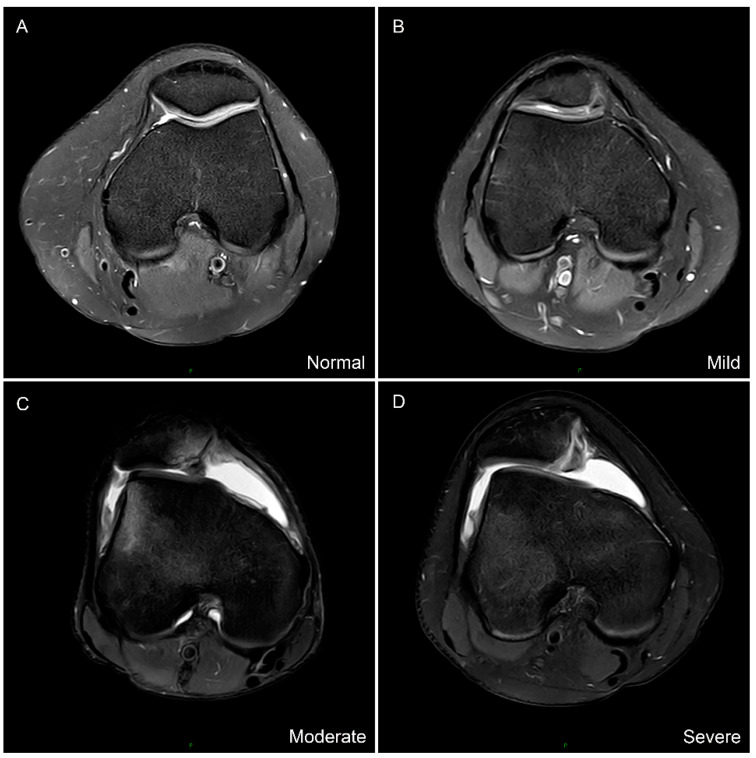
The Oswestry-Bristol classification of trochlear dysplasia. (**A**) Normal femoral trochlea. (**B**) Mild represents shallow trochlea. (**C**) Moderate represents flat trochlea. (**D**) Severe represents convex trochlea.

**Figure 4 jcm-11-05072-f004:**
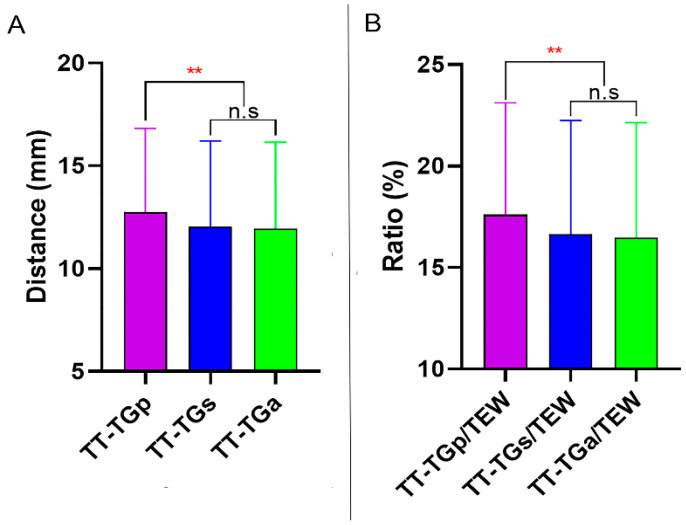
(**A**), Difference among the TT-TG distances in the study group. (**B**) Difference among the TT-TG distances/TEW in the study group. TT-TG distance, tibial tubercle-trochlear groove distance. TEW, surgical transepicondylar width. n.s, none statistically significant. ** Statistically significant with *p* < 0.001.

**Figure 5 jcm-11-05072-f005:**
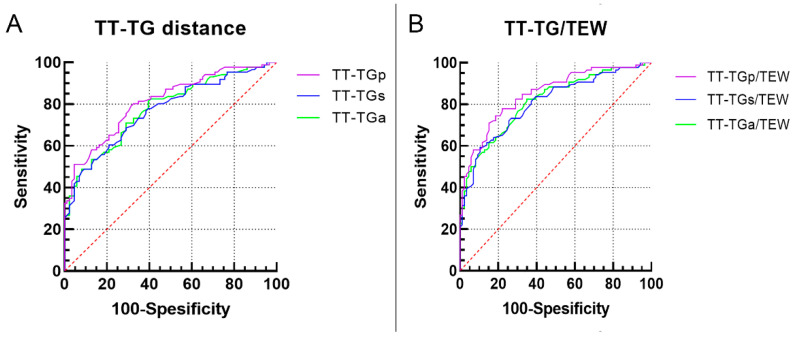
Receiver operating characteristic curves of each tibial tubercle–trochlear groove (TT-TG) distance (**A**) and the three ratios of TT-TG distances/TEW (**B**). TEW, surgical transepicondylar width.

**Figure 6 jcm-11-05072-f006:**
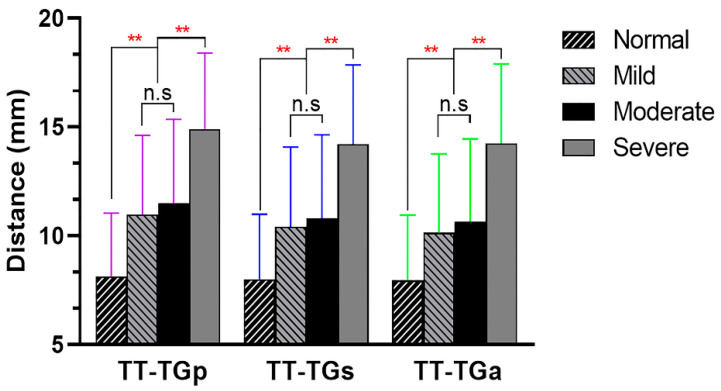
Comparison of the TT-TG distance among different types of trochlear dysplasia (Oswestry-Bristol Classification). n.s, none statistically significant. ** Statistically significant with *p* < 0.001.

**Table 1 jcm-11-05072-t001:** Demographic characteristics of the participants.

Variables	Study Group (n = 86)	Control Group (n = 86)	*p* Value
Sex, n			n.s
Female	66	66	
Male	20	20	
Age, mean ± SD, y	21.2 ± 7.8	22.4 ± 7.2	n.s
Side of knee, n			n.s
Left	40	40	
Right	46	46	

SD, standard deviation; n.s, none statistically significant.

**Table 2 jcm-11-05072-t002:** Differences in the measurements between the study group and the control group.

Variables	Study Group	Control Group	*p* Value
TT-TG distance, mm			
TT-TGp	12.7 ± 4.0	8.2 ± 2.9	<0.001
TT-TGs	12.0 ± 4.1	8.1 ± 3.0	<0.001
TT-TGa	11.9 ± 4.2	8.0 ± 3.0	<0.001
TT-TG/TEW, %			
TT-TGp/TEW	17.6 ± 5.5	10.7 ± 3.8	<0.001
TT-TGs/TEW	16.6 ± 5.6	10.5 ± 3.9	<0.001
TT-TGa/TEW	16.4 ± 5.6	10.4 ± 3.9	<0.001
Trochlear dysplasia (Normal/Mild/Moderate/Severe)	4/24/20/38	84/2/0/0	<0.001 ^a^
PCA, degree	4.0 ± 1.9	3.2 ± 1.6	0.005 ^b^
LPD, mm	20.6 ± 2.8	23.6 ± 2.0	<0.001
MPD, mm	25.4 ± 2.5	25.6 ± 1.7	0.457
TEW, mm	72.3 ± 5.7	76.7 ± 5.3	<0.001
LPD/TEW, %	28.5 ± 3.3	30.9 ± 2.7	<0.001
MPD/TEW, %	35.2 ± 2.8	33.5 ± 2.3	<0.001
LPD/MPD, %	81.3 ± 6.6	92.3 ± 5.2	<0.001

TT-TGp, TT-TGs, and TT-TGa distance, the tibial tubercle to trochlear groove distance with posterior femoral condylar reference line (PCRL), surgical transepicondylar axis (SEA), and anatomical transepicondylar axis (AEA) as reference line, respectively; LPD, lateral posterior femoral condylar distance; MPD, medial posterior femoral condylar distance; TEW, surgical transepicondylar width; PCA, posterior condylar angle; ^a^, the result of chi-square test; ^b^, the result of Wilcoxon rank-sum test; *p* < 0.05 indicates statistically significant.

**Table 3 jcm-11-05072-t003:** The correlations between TT-TG distance parameters and the measurements of femoral condylar morphology, showing the r and ρ value.

	PCA	LPD/MPD	Trochlear Dysplasia ^a^
TT-TGp distance	0.046	0.025	0.474 **
TT-TGs distance	0.045	0.019	0.467 **
TT-TGa distance	0.037	0.039	0.504 **
TT-TGp/TEW	0.037	0.001	0.492 **
TT-TGs/TEW	0.037	−0.004	0.479 **
TT-TGa/TEW	0.026	0.019	0.509 **

TT-TGp, TT-TGs, and TT-TGa distance, the tibial tubercle to trochlear groove distance with posterior femoral condylar reference line (PCRL), surgical transepicondylar axis (SEA), and anatomical transepicondylar axis (AEA) as reference line, respectively; LPD, lateral posterior femoral condylar distance; MPD, medial posterior femoral condylar distance; TEW, surgical transepicondylar width; PCA, posterior condylar angle; ^a^, Oswestry-Bristol Classification; **, ρ value of Spearman correlation analysis showing statistical significance, *p* < 0.001.

**Table 4 jcm-11-05072-t004:** Sensitivity and Specificity of the Cutoff Values.

	AUC	*p* Value	Cutoff Value	Sensitivity, %	Specificity, %
TT-TGp	0.811	<0.001	9.3	80.2	67.4
TT-TGs	0.777	<0.001	8.5	82.6	60
TT-TGa	0.768	<0.001	9.4	68.6	70.9
TT-TGp/TEW	0.85	<0.001	13.9	77.9	77.9
TT-TGs/TEW	0.814	<0.001	11.9	82.6	65.1
TT-TGa/TEW	0.805	<0.001	14.8	59.3	89.5

TT-TGp, TT-TGs, and TT-TGa distance, the tibial tubercle to trochlear groove distance with posterior femoral condylar reference line (PCRL), surgical transepicondylar axis (SEA), and anatomical transepicondylar axis (AEA) as reference line, respectively; TEW, surgical transepicondylar width; AUC, area under the curve, with a value more than 0.7 indicating fair capacity.

**Table 5 jcm-11-05072-t005:** Binary logistic regression model of the TT-TG distance parameters for patellar dislocation.

	Pathological Value	OR	95% CI	*p* Value
TT-TGp	12.9 mm	18.68	6.28, 55.50	<0.001
TT-TGs	13.0 mm	11.12	4.08, 30.23	<0.001
TT-TGa	12.9 mm	14.76	4.96, 43.95	<0.001
TT-TGp/TEW	16.9%	17.78	6.56, 48.21	<0.001
TT-TGs/TEW	16.9%	17.01	5.72, 50.58	<0.001
TT-TGa/TEW	16.8%	11.12	4.09, 30.23	<0.001

TT-TGp, TT-TGs, and TT-TGa distance, the tibial tubercle to trochlear groove distance with posterior femoral condylar reference line (PCRL), surgical transepicondylar axis (SEA), and anatomical transepicondylar axis (AEA) as reference line, respectively; TEW, surgical transepicondylar width; OR, odds ratio; CI, confidence interval.

**Table 6 jcm-11-05072-t006:** Inter-observer reliability of the TT-TG distance in different Oswestry-Bristol Classification of trochlear dysplasia ^a^.

Degree of Trochlear Dysplasia	TT-TGp Distance, mm	ICC (95% CI)	TT-TGs Distance, mm	ICC (95% CI)	TT-TGa Distance, mm	ICC (95% CI)
Normal	8.1 ± 2.8	0.987 (0.977, 0.993)	8.0 ± 2.9	0.984 (0.976, 0.993)	7.9 ± 2.9	0.988 (0.983, 0.992)
Mild	11.0 ± 3.7	0.985 (0.964, 0.996)	10.4 ± 3.8	0.995 (0.988, 0.997)	11.0 ± 3.7	0.987 (0.966, 0.995)
Moderate	11.5 ± 3.9	0.926 (0.875, 0.969)	10.8 ± 3.8	0.923 (0.889, 0.945)	10.6 ± 3.8	0.926 (0.895, 0.951)
Severe	14.9 ± 3.5	0.892 (0.835, 0.938)	14.2 ± 3.6	0.876 (0.841, 0.922)	15.2 ± 3.6	0.883 (0.857, 0.906)

^a^, Distances are expressed as mean ± standard deviation in millimeters. TT-TGp, TT-TGs, and TT-TGa distance, the tibial tubercle to trochlear groove distance with posterior femoral condylar reference line (PCRL), surgical transepicondylar axis (SEA), and anatomical transepicondylar axis (AEA) as reference line, respectively; ICC, intraclass correlation coefficient, with a value more than 0.75 indicating excellent agreement; CI; confidence interval.

## Data Availability

The data associated with the paper are not publicly available but are available from the corresponding author upon reasonable request.

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
