# Peer review of "Tibial Tubercle to Trochlear Groove Distance Measured by Posterior Condylar Reference Line on MRI Can Over-Evaluate Lateralization Deformity of Tibial Tubercle in Patients with Recurrent Patellar Dislocation"

_jcm, 2022, doi:10.3390/jcm11175072_

Round 1

Reviewer 1 Report

L29 : please define tt-tg

l.80 : why only orthopedist when it is a radiologist task ? at least one radiologist should be included amongst the readers, especially to check interobserver agreement

l.80:

 - were the measurements compared to radiographs ? (gold standard)

- did any of the patients had any prior CT for comparison?

l.233: also affected by knee positioning in a MRI coil

The main limitation of this study is the absence of comparison with established imaging techniques, especially with radiographs. Even though the authors mention that measurements might differ from CT, these measures are often realized on CT slices. The absence of comparison with the postoperative outcomes, as mentioned in the limitations section, is also of great concern. Those issues do not allow the extension of this study’s results to the general practice.

Reviewer 2 Report

Thank you for the privilege of reviewing your work on TT-TG distances.

Title: Tibial tubercle to trochlear groove distance measured by transepicondylar axis on MRI can reliably evaluate lateralisation deformity of tibial tubercle in patients with recurrent patella dislocation

Summary:

TT-TG measured with the transepicondylar axis (anatomical or surgical) shows good inter-observer reliability regardless of the trochlear dysplasia. Pathological value of 13.0mm was found however is yet to show any clinical correlation or indication for surgery. TTTGp vs TTTGs/TTTGa distance was 0.7mm and the author is suggesting that the PCRL reference may therefore over-evaluate the TT-TG in PD patients. 

What I find contradicting/difficult is that the TT-TGp has the best diagnostic capacity from the three references and yet the conclusion is that it can overestimate the lateralization, and you recommend using alternatives with worse diagnostic abilities? Please explain further how this conclusion meets your data.

Comments by section:

Abstract:

Line 22-24: need more information/results to justify the conclusion

Introduction 

Well written, with sufficient/extensive background for the paper

Particularly lines 39-51 provide a strong basis for the paper

Materials and methods 

Explained well and extensively. Good figures explaining the methodology clearly.

line 80-83 - only having 2 people measuring the data may lead to biases and given the power dynamic between the senior nad junior orthopedist may result in bias when there are discrepancies. These biases should be in the limitations section.

Results 

Results were well displayed with appropriate graphs and tables.

Discussion

Compared to the rest of the paper the discussion was difficult to follow.

Line 226-237 had quite a few grammatical and language errors that were very awkward. To name a few "most inspiring, "has been attached much importance by lots of researchers". Please revise these two paragraphs to ensure the English is on par with the rest of the paper. 

Paragraph starting on line 238 was confusing. It seems like this concept of condylar dysplasia was much more clearly explained in the introduction paragraph starting from line 39-51. Lines such as the one starting on line 243 were difficult to follow the logic of. Please revise to make the point of condylar dysplasia affecting PCRL more clear, like in the introduction. 

Paragraph starting on 246. More discussion is required for statements such as the one in 248-249 to arise. Please provide more background information or justification. Also line 251-253 made no sense to me. Please revise to relay your point more clearly. 

Line 277- how was the value of 13.0mm found? I could not find the reason why this was the pathological value. Please elaborate.

limitations

- there are more limitations than these please elaborate.

- please elaborate on limitation 3 - line 292 - what do you mean by limited?

- a good study should clearly explain and outline their limitations.

Conclusion

This conclusion is far too short. It should clearly state the main finding which is the reliability of the transepicondylar axis and explain briefly why the TT-TGp over-estimates. (eg "condylar dysplasia may be a reason for the overestimating of the ..."). Additionally, it should be clear in the conclusion that this has not been correlated to clinical outcomes and further studies are required
